# Loop Diuretic Dose and Nutritional Status of Patients with Heart Failure with Reduced Ejection Fraction

**DOI:** 10.3390/nu17132105

**Published:** 2025-06-25

**Authors:** Filip Sawczak, Aleksandra Soloch, Maria Cierzniak, Alicja Szubarga, Kamila Kurkiewicz-Sawczak, Agata Kukfisz, Magdalena Szczechla, Helena Krysztofiak, Magdalena Dudek, Ewa Straburzyńska-Migaj, Marta Kałużna-Oleksy

**Affiliations:** 11st Department of Cardiology, Poznan University of Medical Sciences, Dluga 1/2, 61-848 Poznan, Poland; fsawczak@gmail.com (F.S.); ola.soloch@gmail.com (A.S.);; 2Department of Paediatric Cardiology, Poznan University of Medical Sciences, Szpitalna 27/33, 60-572 Poznan, Poland; kkurkiewicz75@gmail.com; 33rd Department of Cardiology, Silesian Center for Heart Diseases, Medical University of Silesia, 41-800 Zabrze, Poland

**Keywords:** BMI, CONUT, furosemide, GNRI, heart failure, MNA, malnutrition, nutrition, nutritional risk, torsemide

## Abstract

**Background/Objectives:** Loop diuretics are among the most commonly used drugs in patients with heart failure with reduced ejection fraction (HFrEF). Higher doses of these diuretics are associated with poorer clinical status and may contribute to malnutrition. The study aims to assess the relationship between the use of high-dose loop diuretics and nutritional status in patients with HFrEF. **Methods:** The study included 353 hospitalized patients with HFrEF. Nutritional status was assessed using the Mini Nutritional Assessment (MNA), the Geriatric Nutritional Index (GNRI), and the CONtrolling NUTritional status (CONUT). Patients were divided into three groups according to the daily dose of loop diuretics (defined as furosemide equivalent = 1 × furosemide dose and 2 × torsemide dose): low dose (LD), 40 mg/day or no treatment; medium dose (MD), 41–160 mg/day; or high dose (HD), >160 mg/day. **Results:** Of the evaluated patients, the mean MNA score was 23.31 ± 2.93 points, and 49.8% were at risk of malnutrition or malnourished. According to the MNA, patients in HD and MD groups had worse nutritional status than the LD group, similarly according to the GNRI. For CONUT, the differences were significant between all groups: nutritional status was the worst in the HD group, intermediate in the MD group, and the best in the LD group. **Conclusions:** The intake of loop diuretics, especially in high doses, correlates with an elevated risk of malnutrition in patients with HFrEF independently of sex, age, NYHA class, and left ventricular ejection fraction.

## 1. Introduction

The nutritional status of patients suffering from heart failure (HF) is associated with both quality of life and prognosis. Impaired nutritional status can be present in 10–90% of HF patients, depending on the studied group and the tool used for assessment [1,2,3,4,5,6,7]. Malnutrition is among the numerous factors that adversely affect the prognosis and reduce life expectancy [8,9]. Therefore, it is important to identify possible factors associated with poor nutritional status of patients with HF, including medications commonly used in this population. Identification of the most vulnerable patients could help to address their needs with nutritional support, including dietitian consultation. Numerous studies emphasize the importance of taking care of nutritional status in HF. Patients should avoid salt [10] and must not drink alcohol. Optimally, a balanced diet—Mediterranean [10,11,12] or DASH [10,13,14]—should be followed. Additionally, there are data that supplementation with certain components may be beneficial: middle-chain triglyceride oil, beta-hydroxybutyrate salts, ketone esters, and coenzyme Q10 [12]. Supplementation with these components is thought to be cardioprotective, possibly due to an increase in myocardial energy production. Additionally, there are reports that supplementation with thiamine, ubiquinol, D-ribose, and L-arginine could enhance left ventricular ejection fraction [12]. Amino acid or protein supplementation may decrease sarcopenia and improve aerobic capacity [15,16,17].

Loop diuretics are commonly prescribed in patients with heart failure to relieve symptoms of congestion and maintain euvolemia. According to the 2021 European Society of Cardiology (ESC) guidelines, loop diuretics should reduce congestion symptoms in patients with HF, regardless of ejection fraction and etiology [18]. In heart failure patients, the use of loop diuretics—especially at higher doses—is often linked to persistent clinical congestion, a factor associated with a worse prognosis [19]. Higher diuretic doses are often associated with a worse clinical status, as the need for intensified diuretic therapy typically reflects more advanced disease severity. Notably, in the study by Pellicori et al., multivariable analysis demonstrated that markers of heart failure progression, rather than loop diuretic use itself, were predictive of prognosis [19]. Thus, the use of loop diuretics should be considered a marker—rather than a direct contributor—to poorer outcomes in heart failure patients.

The occurrence of adverse effects, such as dyselectrolytemia and taste disturbances, is also linked to the overall clinical condition of heart failure patients. Taste disturbances are observed more frequently in individuals treated with furosemide [20], potentially due to changes in salivary composition that directly influence taste perception [21]. These alterations may further impact patients’ nutritional status. Some patients with HFrEF do not require high doses of diuretics, and their doses can be carefully reduced under regular monitoring of euvolemia [22]. Usually, higher doses are given during the hospitalization or close after the discharge from the hospital (de novo diagnosis, exacerbation of HF) and then reduced. As loop diuretics could cause a metallic taste as one of their side effects, reducing the dose of diuretics may theoretically improve appetite and consequently the nutritional status.

Given the substantial impact of nutritional status on prognosis and the widespread use of loop diuretics in heart failure management, it is important to explore the relationship between diuretic use and patients’ nutritional status. The study results could help select especially vulnerable patients who should undergo more frequent monitoring of nutritional status and nutritional interventions.

The study aimed to examine differences in nutritional status between groups of HF with reduced ejection fraction (HFrEF) patients with low, medium, and high doses of loop diuretics. The purpose was to assess if there is a correlation between the dosage of loop diuretics and nutritional status.

## 2. Materials and Methods

This prospective cohort study included 353 consecutive patients with chronic HFrEF who were admitted to the department of cardiology at the university hospital between January 2019 and December 2022. Inclusion criteria were as follows: (1) 18 years or older; (2) HF diagnosis according to the International Classification of Diseases (ICD-10 code for primary diagnosis I50); (3) HF duration longer than 3 months; (4) left ventricular ejection fraction (LVEF) ≤ 40%; (5) signed informed consent form to participate in the study. We enrolled both stable patients admitted for HF assessment and patients admitted due to exacerbation of HF. Patients with decompensation of HF were enrolled after stabilization of their clinical status. The exclusion criteria were HF de novo and age under 18 years old, and inability to give informed consent. The enrollment process is presented in Figure 1. The present research was conducted in accordance with the Declaration of Helsinki and was approved by the Ethics Committee of Poznan University of Medical Sciences (approval code 447/19). All patients were informed and agreed to participate in the study.

Clinical history, laboratory parameters, and prescribed medications were prospectively recorded. New York Heart Association (NYHA) classification was used to assess the clinical stage of the disease according to the ESC guidelines [18]. Clinical data included age, body mass index (BMI), and the cause of HF (ischemic or non-ischemic). BMI was calculated with the following formula: BMI = weight (kg)/[height (m)^2^] [23]. In addition, electrocardiography and echocardiography were performed, and the Simpson method was used to assess LVEF [24]. Laboratory parameters included complete blood count, B-type natriuretic peptide (BNP), *N*-terminal prohormone of B-type natriuretic peptide (NT-proBNP), creatinine, lipid profile, thyroid-stimulating hormone, sodium, potassium, *C*-reactive protein, albumin, and total protein. The blood tests were performed upon admission. The Modification of the Diet in Renal Disease (MDRD) formula was used to count the estimated glomerular filtration rate (eGFR) [25].

Nutritional status was assessed using the Mini Nutritional Assessment (MNA) questionnaire, as well as the Geriatric Nutritional Index (GNRI) and the CONtrolling NUTritional status (CONUT). The MNA form is an 18-question form divided into screening (6 questions) and assessment (12 questions) [26]. Calf and mid-arm circumferences and BMI must be measured. BMI cut-off values are <19 kg/m^2^, 19 kg/m^2^ ≤ BMI < 21 kg/m^2^, 21 kg/m^2^ ≤ BMI < 23 kg/m^2^, and BMI ≥ 23 kg/m^2^. These correspond to 0, 1, 2, and 3 points. Patients can receive 30 points: 14 at screening and 16 at assessment. The scores are divided into three groups: normal nutritional status (24–30 points), at risk of malnutrition (17–23.5 points), and malnourished (less than 17 points). The MNA concentrates mostly on self-reported nutritional habits and daily living activities [26]. GNRI scores are based on BMI and albumin concentration. GNRI is calculated with the following formula: [1.489 × serum albumin (g/L)] + [41.7 × body weight/ideal body weight (kg)] [27]. Ideal body weight is assessed using the following formula: ideal body weight = height^2^ (m) × 22.40. If the body weight/ideal body weight is > 1, the ratio is set to 1. A GNRI score > 98 indicates that the subject is a patient with no nutritional risk, and ≤98 indicates a risk of malnutrition [27,28]. CONUT score is based on the laboratory parameters [29]. Patients receive 0–3 points depending on cholesterol level (lower—more points), 0–3 points depending on lymphocyte concentration (lower lymphocytes—more points), and 0–6 points depending on albumin level (lower albumin—more points), and then the points are summed up [29]. The more points, the worse the nutritional status is. Patients with two or more points were classified as at risk of malnutrition [29].

Daily doses of loop diuretics were measured as furosemide equivalent = 1 × furosemide dose (mg/day) and 2 × torsemide dose (mg/day). Patients received diuretics at the studied doses for at least one month. Patients were divided into three groups according to the daily dose of loop diuretics: low dose (LD), ≤40 mg/day or no loop diuretic treatment; medium dose (MD), 41–160 mg/day; or high dose (HD), >160 mg/day. The MNA score and CONUT and variables without normal distribution were compared using the Kruskal–Wallis ANOVA test with post hoc tests. GNRI and normally distributed variables were compared between groups with ANOVA and post hoc Tukey range tests. Categorical variables were compared with Chi2 tests. Subsequently, to minimize the impact of the worse clinical status of HD patients on the results, one LD and one MD control were matched for each HD patient. It was performed with propensity score matching using the nearest neighbor with the limit method. The propensity score was counted with logistic regression analysis from sex, age, left ventricular ejection fraction, and symptoms of HF in NYHA class III or IV. Then, matched groups of LD, MD, and HD patients were compared with ANOVA and Kruskal–Wallis ANOVA with post hoc tests, the same as the unmatched population. *p* < 0.05 was considered significant for all tests. Statistical analysis was performed with Statistica 13.3 (Tibco Software Inc., Palo Alto, CA, USA).

The primary endpoint was the difference in MNA score between the low-, medium-, and high-dose loop diuretic groups. Differences in GNRI and CONUT between groups on low, medium, and high doses of loop diuretics were secondary endpoints. Differences in MNA score, GNRI, and CONUT between patients using high diuretic doses and matched groups of patients taking low and medium doses were additional secondary endpoints.

## 3. Results

The study enrolled 353 patients with HFrEF, including 17% females. The mean age was 55 ± 12 years (minimum 20, maximum 85 years), and the mean LVEF was 25 ± 8%. About half of the patients had ischemic etiology of HF. Most of the patients took loop diuretics (92.4%), beta-blockers (98.6%), mineralocorticoid receptor antagonists (MRA, 87.8%), and angiotensin-converting enzyme inhibitors (ACEI), angiotensin receptor blockers (ARB), or neprilysin inhibitors (ARNI) (89.5%). Sodium–glucose transport protein 2 (SGLT2) inhibitors were less commonly prescribed due to the time of enrollment (19.5%).

A total of 42.5% of patients were treated with both loop diuretics (torsemide and furosemide), 36.3% received only torsemide, and 13.6% only furosemide. Most of the patients (43.1%) were in the LD group, and 17.8% of LD patients (7.7% of the whole group) were without loop diuretic treatment. The median furosemide equivalent in the whole enrolled group was 80 (IQR: 20–160) mg/day, and in the HD group 280 (IQR: 220–390) mg/day. General characteristics of the full studied group and comparison according to the loop diuretic doses are presented in Table 1.

The mean MNA score was 23.31 ± 2.93; 2.8% of patients were malnourished, and 47% were at risk of malnutrition. The mean GNRI was 101.96 ± 8.7, and accordingly, 27.1% of patients were at risk of malnutrition. The CONUT score was 2.36 ± 2.13 on average, accounting for 57.4% of the study population at nutritional risk.

Results on the nutritional status are mostly consistent regardless of the tool used (Figure 2). According to the MNA questionnaire, patients taking high or medium diuretic doses had worse nutritional status than the low-dose group (HD vs. LD, *p* < 0.001; MD vs. LD, *p* = 0.001). Similarly, nutritional status was worse in the MD and HD groups than in the LD group according to the GNRI (HD vs. LD, *p* < 0.001; MD vs. LD, *p* = 0.017). For CONUT, the differences were significant between all groups: nutritional status was the worst in the HD group, intermediate in the MD group, and the best in the LD group (HD vs. LD, *p* < 0.001; MD vs. LD, *p* = 0.032; MD vs. HD, *p* = 0.005). BMI was higher in the HD group compared with the LD and MD groups (HD vs. LD, *p* = 0.011; HD vs. MD, *p* = 0.028). A post hoc comparison of MNA score, GNRI, CONUT, and BMI between LD, MD, and HD was presented in Table 2.

Patients receiving higher doses of loop diuretics were characterized by older age, more advanced NYHA class, more common AF and IHD etiology, as well as lower GFR. Moreover, they had lower levels of albumin, total cholesterol, low-density lipoprotein cholesterol, high-density lipoprotein cholesterol, triglycerides, hemoglobin, sodium, and potassium. In addition, we observed higher BNP, NT-proBNP, and *C*-reactive protein in this group. Compared to the MD and LD groups, fewer patients from the HD group took SGLT2 and ARNI.

Propensity score matching was performed with the logistic regression method using age, sex, LVEF, and NYHA class III or IV as predictors of high doses of diuretics. Appropriate LD and MD controls were found for 55 of 87 HD patients. Nutritional status according to MNA and GNRI was worse in the HD group compared with the LD group (MNA: HD vs. LD—*p* = 0.012, GNRI: HD vs. LD—*p* = 0.042). A comparison of the nutritional status of patients matched with propensity score according to the loop diuretic dose is presented in Figure 3 and Table 3.

## 4. Discussion

The study demonstrated that higher doses of loop diuretics are associated with worse nutritional status according to the MNA, GNRI, and CONUT. Importantly, the results are cohesive across all tools used. Propensity score matching successfully allowed comparing patients with different doses of loop diuretics and no differences in age, sex, LVEF, and NYHA class. Nutritional status was still worse in the HD group. It indicates that a high dose of loop diuretics is a risk factor for worse nutritional status independently of sex, age, LVEF, and NYHA.

Previous research on this issue is limited. We did not find any article concerning the association of nutritional status with loop diuretic dosing or even usage in HF patients. Some studies focus on prognosis by comparing groups with different nutritional status [30,31,32]. In addition to the clinical outcomes, they compared different patients’ characteristics, including the use of loop diuretics.

The study of Chen et al. retrospectively enrolled 371 patients and compared groups of patients with different nutritional risks according to CONUT [30]. It concentrated on the prognosis; however, it showed that more patients with a higher risk of malnutrition take loop diuretics. The study of Abulmiti et al. included patients with acute decompensated HF, concentrating on the association of Chanye–Stokes breathing and malnutrition presence on prognosis [31]. Additionally, it showed that malnourished patients had more often taken loop diuretics compared to those without malnutrition. Wu et al.’s study comprised 108 patients with decompensated severe systolic heart failure [32]. It also concentrated on the association of nutritional status with prognosis. It employed a Nutritional Risk Index—a tool similar to the GNRI used in our study. Apart from the association with prognosis, it showed that loop diuretics are more frequently used in patients with NRI ≤93—with worse nutritional status compared to those with better nutritional status. Moreover, it analyzed the dose of diuretics, with a mean dose of furosemide 55 ± 28 mg in patients with worse nutritional status and 38 ± 28 in patients with better nutritional status. However, it is not mentioned if furosemide was the only loop diuretic taken by patients and if the others were omitted, recalculated, or not applied, which is a limitation [32].

Patients receiving higher doses of loop diuretics were older and had more common comorbidities. It could influence nutritional status, confounding the study results. However, propensity score matching was performed to assess whether differences in nutritional status were independent of these confounders. In matched groups of LD, MD, and HD patients (Table 3), no differences in age and comorbidities, except for GFR, were observed.

Loop diuretics use can cause a deficit of nutrients in two ways: it may cause a loss of electrolytes with urine [33,34,35] and affect appetite and therefore increase the risk of malnutrition. Loop diuretics cause a substantial loss of both magnesium and potassium in the plasma and intracellular space [33,34]. Increasing doses of diuretics can compromise renal perfusion, stimulating the renin–angiotensin–aldosterone system [35]. Simultaneous activation of vasopressin can cause retention of free water, leading to hyponatremia [35]. On the other hand, diuretics can lead to decreased appetite by changing the composition of saliva. Impaired appetite can lead to reduced food intake and, consequently, malnutrition, causing a deficiency of various nutrients [36].

Our study is the first dedicated to analyzing the association of different doses of loop diuretics in HF with nutritional status. Furthermore, our study employed propensity score matching to show a more transparent view of this association. Moreover, only one study assessed the dose of loop diuretic; however, it took into consideration only furosemide without further clarification. The analysis of diuretics appears to be vital, as the vast majority of HF patients take them. Additionally, incorporating a few methods of malnutrition assessment, combining biochemical and clinical parameters such as CONUT and GNRI, and patient-reported criteria as MNA, shows the coincidence of high loop diuretic doses and worse nutritional status assessed with multiple tools. Therefore, patients with high doses of diuretics should undergo frequent monitoring of their nutritional status and, if necessary, nutritional interventions. Previous research indicates that loop diuretic doses can be reduced in stable HFrEF patients [22]. Our results may indicate that such a reduction in loop diuretic use could have a beneficial effect on nutritional status; however, it requires further experimental studies.

### Study Limitations

The study is one-center and observational; thus, it cannot define a causative relationship on its basis. Numerous confounders, like HF advancement, could have affected the results; however, propensity score matching was performed to limit the effect of confounding factors. Moreover, experimental, randomized studies with different diuretic doses would be ethically unacceptable due to the elementary need for decongestion of HF patients. Therefore, observational studies are essential for assessing the loop diuretic impact on HF patients. The share of female patients was much lower than that of male patients, which is related to the relatively young HFrEF patients. Body composition could further explain nutritional differences between patients, but it was not assessed in this study. However, using three different methods—MNA, GNRI, and CONUT—provides more reliable results. We did not gather information about the place of residence of the patients—rural or urban—or the time from HF diagnosis for all patients enrolled in this study and it was not analyzed. This is a cross-sectional study; consequently, we do not gather data on patient prognosis.

## 5. Conclusions

The intake of loop diuretics, especially in high doses, correlates with an elevated risk of malnutrition in patients with HFrEF independently of sex, age, NYHA class, and LVEF. Patients receiving loop diuretics, especially high doses, should be screened for malnutrition and receive special nutritional support.

## Figures and Tables

**Figure 1 nutrients-17-02105-f001:**
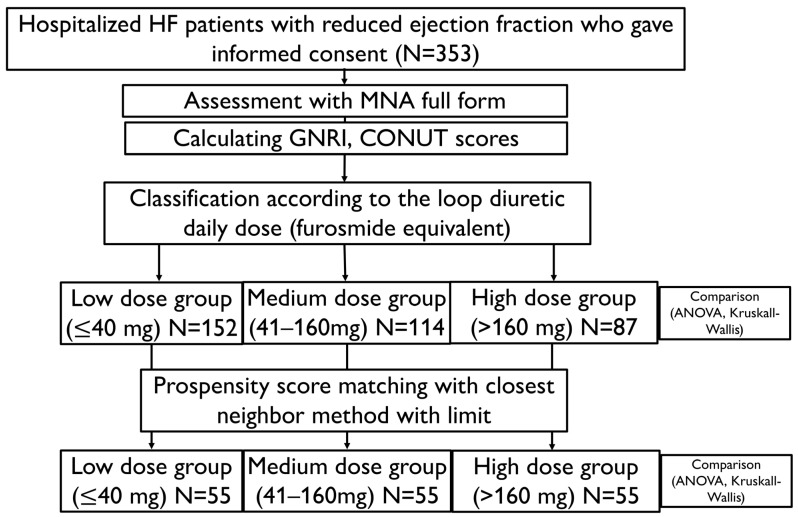
Flowchart visualizing the enrollment process and analysis. CONUT—Controlling Nutritional Status (score); GNRI—Geriatric Nutritional Risk Index; HF—heart failure; MNA—Mini Nutritional Assessment.

**Figure 2 nutrients-17-02105-f002:**
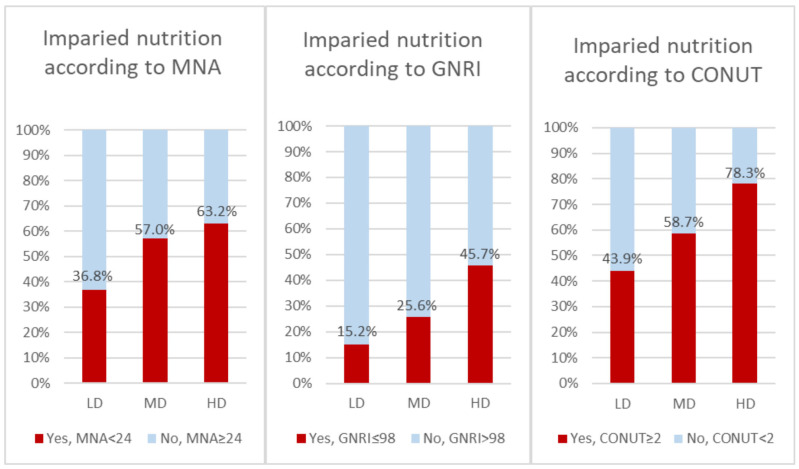
Comparison of nutritional status depending on loop diuretic dose in whole group (*n* = 353). CONUT—Controlling Nutritional Status (score); GNRI—Geriatric Nutritional Risk Index; MNA—Mini Nutritional Assessment.

**Figure 3 nutrients-17-02105-f003:**
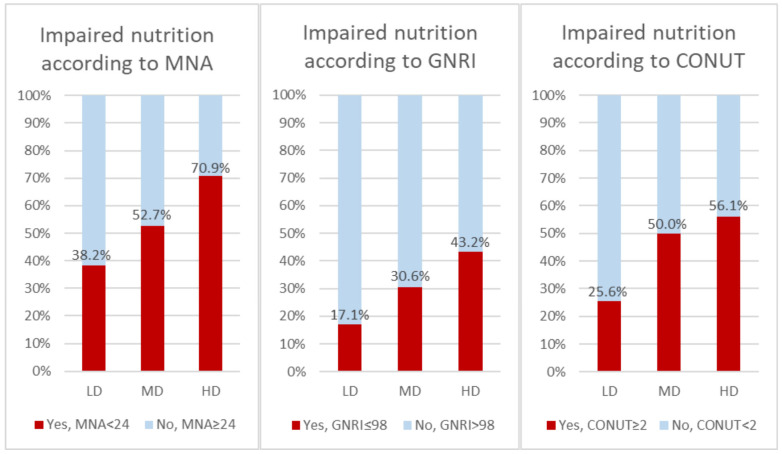
Comparison of nutritional status depending on loop diuretic dose in groups matched with propensity score (*n* = 165). CONUT—Controlling Nutritional Status (score); GNRI—Geriatric Nutritional Risk Index; MNA—Mini Nutritional Assessment.

**Table 1 nutrients-17-02105-t001:** General characteristics of the full studied group and comparison according to the loop diuretic doses.

Characteristic	All (*n* = 353)	Low-Dose Diuretics (*n* = 152)	Medium-Dose Diuretics (*n* = 114)	High-Dose Diuretics (*n* = 87)	*p*
Age (years)	55.37 ± 12.12	52.63 ± 13.54	56.94 ± 10.41	58.11 ± 10.65	0.001
Female sex	60 (17.0%)	30 (19.7%)	24 (21.1%)	6 (6.9%)	0.015
IHD etiology	173 (49.0%)	63 (41.7%)	60 (53.1%)	50 (57.5%)	0.040
BMI (kg/m^2^)	26.68 ± 5.28	28.13 ± 4.92	28.26 ± 5.39	30.17 ± 5.52	0.010
LVEF (%)	24.58 ± 7.90	27.51 ± 8.01	22.02 ± 6.88	22.89 ± 7.33	<0.001
Nutritional assessment					
MNA score	23.31 ± 2.93	24.06 ± 2.94	22.96 ± 2.92	22.47 ± 2.64	<0.001
MNA score ≥ 24	177 (50.1%)	96 (63.2%)	49 (43.0%)	32 (36.8%)	<0.001
MNA score 17–23.5	166 (47.0%)	51 (33.6%)	62 (54.4%)	53 (60.9%)	<0.001
MNA score <17	10 (2.8%)	5 (3.3%)	3 (2.6%)	2 (2.3%)	0.895
GNRI score	101.96 ± 8.70 {102} *	104.66 ± 9.43 {53}	101.51 ± 7.13 {32}	98.66 ± 8.15 {17}	<0.001
GNRI score ≤ 98	68 (27.1%) {102}	15 (15.2%) {53}	21 (25.6%) {32}	32 (45.7%) {17}	<0.001
CONUT score	2.36 ± 2.13 {90}	1.68 ± 1.77 {38}	2.31 ± 1.89 {34}	3.55 ± 2.42 {18}	<0.001
CONUT score ≥ 2	151 (57.4%) {90}	50 (43.9%) {38}	47 (58.7%) {34}	54 (78.3%) {18}	<0.001
Comorbidities					
DM	100 (28.3%)	34 (22.4%)	35 (30.7%)	31 (35.6%)	0.078
COPD	29 (8.2%)	6 (3.9%)	16 (14.0%)	7 (8.0%)	0.013
Chronic renal insufficiency GFR MDRD < 60 mL/m^2^/min	129 (36.5%)	33 (21.7%)	45 (39.5%)	51 (58.6%)	<0.001
Arterial hypertension	173 (49.0%)	71 (46.7%)	59 (51.8%)	43 (49.4%)	0.715
Atrial fibrillation	142 (40.2%)	42 (27.6%)	48 (42.1%)	52 (59.8%)	<0.001
NYHA class					
I	6 (1.7%)	6 (3.9%)	0	0	<0.001
II	160 (45.3%)	109 (71.7%)	34 (29.8%)	17 (19.5%)	
III	156 (44.2%)	36 (23.7%)	68 (59.6%)	52 (59.8%)	
IV	31 (8.8%)	1 (0.7%)	12 (10.5%)	18 (20.7%)	
NYHA class III or IV	187 (53.0%)	37 (24.4%)	80 (70.1%)	70 (80.5%)	<0.001
Biochemical parameters					
Hgb (mmol/L)	8.98 ± 1.15	9.14 ± 0.93	8.93 ± 1.22	8.73 ± 1.35	0.025
BNP (pg/mL)	462.6 (209.6–869.7) {163}	238.4 (140.2–578.3) {87}	530.8 (278–815.1) {48}	690.4 (343.5–1290) {28}	<0.001
NT-proBNP (pg/mL)	2231 (867–5054) {88}	1132 (393–2668) {27}	3042 (1421–5062) {29}	5245 (2324–10951) {32}	<0.001
Creatinine (µmol/L)	111.22 ± 49.60	99.10 ± 55.64	108.89 ± 23.32	135.66 ± 44.98	<0.001
eGFR MDRD (mL/min/1.73 m^2^)	69.23 ± 23.91	78.07 ± 22.68	67.93 ± 23.32	55.28 ± 19.61	<0.001
TSH (mIU/L)	1.74 (1.12–2.94) {27}	1.52 (1.06–2.44) {12}	1.70 (1.03–3.12) {9}	2.46 (1.26–3.80) {6}	0.002
Na^+^ (mmol/L)	139.28 ± 3.61	140.86 ± 2.59	138.84 ± 3.28	137.08 ± 4.21	<0.001
K^+^ (mmol/L)	4.35 ± 0.46	4.46 ± 0.40	4.38 ± 0.42	4.14 ± 0.53	<0.001
CRP (mg/L)	4.0 (3.3–8.0) {11}	4.0 (2.8–4.0) {3}	4.0 (3.3–10.0) {4}	7.5 (4.0–15.1) {4}	<0.001
CholT (mmol/L)	4.22 ± 1.29 {39}	4.54 ± 1.25 {11}	4.30 ± 1.31 {17}	3.54 ± 1.08 {11}	<0.001
LDL (mmol/L)	2.47 ± 1.10 {39}	2.74 ± 1.07 {11}	2.49 ± 1.14 {17}	1.95 ± 0.93 {11}	<0.001
HDL (mmol/L)	1.22 ± 0.41 {39}	1.29 ± 0.41 {11}	1.20 ± 0.40 {17}	1.14 ± 0.38 {11}	0.023
Triglycerides (mmol/L)	1.63 ± 1.02 {39}	1.75 ± 1.05 {11}	1.71 ± 1.11 {17}	1.30 ± 0.77 {11}	0.005
Albumin	43.25 ± 11.11 {77}	46.69 ± 12.75 {38}	41.03 ± 8.09 {29}	40.53 ± 10.08 {12}	<0.001
Total protein	69.72 ± 11.34 {64}	68.87 ± 11.74 {30}	70.87 ± 10.67 {24}	69.72 ± 11.46 {10}	0.447
Medications					
Loop diuretics (%)	326 (92.4%)	125 (81.7%)	114 (100%)	87 (100%)	<0.001
Furosemide equivalent (mg)	80 (20–160)	20 (5–40)	100 (80–120)	280 (220–390)	<0.001
Torsemide only (%)	128 (36.3%)	107 (70.4%)	14 (12.3%)	7 (8.0%)	<0.001
Furosemide only (%)	48 (13.6%)	18 (11.8%)	30 (26.3%)	0	<0.001
Both torsemide and furosemide (%)	150 (42.5%)	0	70 (61.4%)	80 (92.0%)	<0.001
β-blocker (%)	348 (98.6%)	149 (98.0%)	114 (100%)	85 (97.7%)	0.289
ACEI/ARB (%)	205 (58.1%)	78 (51.3%)	81 (71.1%)	46 (52.9%)	0.003
ARNI (%)	111 (31.4%)	64 (42.1%)	25 (21.9%)	22 (25.3%)	<0.001
MRA (%)	310 (87.8%)	131 (86.2%)	100 (87.7%)	79 (90.8%)	0.543
Statin (%)	230 (65.2%)	100 (65.8%)	81 (71.1%)	49 (56.3%)	0.087
SGLT-2 inhibitor (%)	69 (19.5%)	40 (26.3%)	18 (15.8%)	11 (12.6%)	0.017

*—numbers in curly brackets; {} indicate number of data lacks for each variable; ACEI—angiotensin-converting enzyme inhibitor; ARB—angiotensin II receptor blocker; ARNI—angiotensin receptor–neprilysin inhibitor; BMI—body mass index; BNP—B-type natriuretic peptide; CholT—total cholesterol; CONUT—Controlling Nutritional Status (score); COPD—chronic obstructive pulmonary disease; CRP—*C*-reactive protein; DM—diabetes mellitus; GFR—glomerular filtration rate; GNRI—Geriatric Nutritional Risk Index; Hgb—hemoglobin; HDL—high-density lipoprotein; IHD—ischemic heart disease; LDL—low-density lipoprotein; LVEF—left ventricular ejection fraction; MDRD—Modification of Diet in Renal Disease (formula); MNA—Mini Nutritional Assessment; MRA—mineralocorticoid receptor antagonist; NT-proBNP—*N*-terminal pro-B-type natriuretic peptide; NYHA—New York Heart Association (functional classification); SGLT-2 inhibitor—sodium–glucose cotransporter-2 inhibitor; TSH—thyroid-stimulating hormone.

**Table 2 nutrients-17-02105-t002:** Detailed comparison of MNA score, GNRI, and BMI between low-, medium-, and high-dose patients before and after PSM; *p* value for post hoc tests.

Before PSM	HD (*n* = 87) vs. LD (*n* = 152)	MD (*n* = 114) vs. LD (*n* = 152)	HD (*n* = 87) vs. MD (*n* = 114)
MNA score	<0.001	0.001	0.457
GNRI	<0.001	0.017	0.146
CONUT score	<0.001	0.032	0.005
BMI	0.011	0.980	0.028
**After PSM**	**HD (*n* = 55) vs. LD (*n* = 55)**	**MD (*n* = 55) vs. LD (*n* = 55)**	**HD (*n* = 55) vs. MD (*n* = 55)**
MNA score	0.012	0.791	0.233
GNRI	0.042	0.332	1.000
CONUT score	0.095	0.288	1.000
BMI	0.073	0.591	0.446

Abbreviations: see Table 1.

**Table 3 nutrients-17-02105-t003:** Comparison of nutritional status of patients matched with propensity score according to the loop diuretic dose.

Characteristic	Low-Dose Loop Diuretics (*n* = 55)	Medium-Dose Loop Diuretics (*n* = 55)	High-Dose Loop Diuretics (*n* = 55)	*p*
Age (years)	59.20 ± 11.67	56.80 ± 11.63	57.29 ± 11.96	0.528
Female sex (%)	8 (14.6%)	3 (5.5%)	5 (9.1%)	0.268
IHD etiology	32 (59.3%)	26 (47.3%)	34 (61.8%)	0.261
LVEF (%)	24.06 ± 8.16	23.17 ± 8.10	24.12 ± 7.42	0.780
Nutritional assessment				
MNA score	23.40 ± 3.60	23.02 ± 3.00	22.26 ± 2.41	0.014
MNA score ≥ 24	34 (61.8%)	26 (47.3%)	16 (29.1%)	0.003
MNA score 17–23.5	16 (29.1%)	27 (49.1%)	39 (70.9%)	<0.001
MNA score < 17	5 (9.1%)	2 (3.6%)	0	0.058
Body mass index	27.72 ± 4.35	28.74 ± 5.71	30.00 ± 6.12	0.093
GNRI score	102.90 ± 8.53 {20} *	100.59 ± 8.03 {19}	98.72 ± 8.52 {11}	0.046
GNRI score ≤ 98	6 (17.1%) {20}	11 (30.6%) {19}	19 (43.2%) {11}	0.046
CONUT score	1.282 ± 1.701 {16}	1.688 ± 1.469	1.976 ± 1.877	0.065
CONUT score ≥ 2	10 (25.6%) {16}	16 (50%) {23}	41 (56.1%) {14}	0.016
NYHA class				
I	1 (1.8%)	0	0	0.115
II	17 (30.9%)	16 (29.1%)	17 (30.9%)	
III	36 (65.5%)	33 (60%)	28 (50.9%)	
IV	1 (1.8%)	6 (10.9%)	10 (18.2%)	
NYHA class III or IV	37 (67.3%)	39 (70.9%)	38 (69.1%)	0.918
Comorbidities				
DM	16 (29.1%)	17 (30.9%)	19 (34.6%)	0.822
COPD	5 (9.1%)	11 (20.0%)	6 (10.9%)	0.197
Chronic renal insufficiency with GFR < 30	1 (1.8%)	1 (1.8%)	3 (5.4%)	0.438
Arterial hypertension	31 (56.4%)	28 (50.9%)	28 (50.9%)	0.803
Atrial fibrillation	21 (38.2%)	26 (47.3%)	33 (60.0%)	0.071
Loop diuretic use				
Furosemide equivalent (mg)	20 (10–40)	100 (80–120)	280 (220–380)	<0.001
Torsemide only (%)	40 (72.7%)	8 (14.5%)	4 (7.3%)	<0.001
Furosemide only (%)	7 (12.7%)	14 (25.5%)	0	<0.001
Torsemide + Furosemide (%)	0	33 (60%)	51 (92.7%)	<0.001
Laboratory parameters				
HGB	9.13 ± 0.99	9.01 ± 1.27	8.79 ± 1.44	0.349
BNP (pg/mL)	249.6 (166.4–649.3) {28}	498.5 (213.1–1095.7) {21}	719.2 (343.5–1285.4) {17}	<0.001
NT-proBNP (pg/mL)	1798 (1049–3378) {14}	2479 (1601–5062) {13}	3622 (2144–13235) {20}	0.003
GFR MDRD (mL/min/1.73 m^2^)	70.50 ± 22.58	66.57 ± 23.21	55.82 ± 19.41	<0.001

*—numbers in curly brackets; {} indicate number of data lacks for each variable; Abbreviations: see Table 1.

## Data Availability

The original contributions presented in this study are included in the article. Further inquiries can be directed to the corresponding author.

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
