# Peer review of "Loop Diuretic Dose and Nutritional Status of Patients with Heart Failure with Reduced Ejection Fraction"

_nutrients, 2025, doi:10.3390/nu17132105_

Round 1
Reviewer 1 Report
Comments and Suggestions for Authors
First of al I congratulate the authors for their work. The study is very interesting and assesses the association between nutritional status and loop diuretics.
Here some suggestions:
Material and methods section
-Mention the exclusion criteria applied
-The person who performed the ecocardiography was the same for all patients?
-at what point during hospitalization blood tests were taken
-it would have been interesting to consider the background - rural or urban
-it would have been interesting to consider the length of time since they have been diagnosed with heart failure
-how long they were treated with diuretics at the doses studied
Discussion section
-comment the fact that patients with higher doses of loop diuretics were older, had comorbities (more common AF and DM, and lower GFR) which can influent nutritional status
Author Response
We are grateful to the Reviewers for the comments which helped us improve the manuscript. We have modified our manuscript in accordance with the recommendations of the Reviewers. We hope that now our publication will be unambiguous and more understandable to the reader.
First of all I congratulate the authors for their work. The study is very interesting and assesses the association between nutritional status and loop diuretics.
Here some suggestions:
Material and methods section
Comment 1. - Mention the exclusion criteria applied
​​Response 1. We excluded patients with HF de novo, under 18 years old or unable to gave informed consent. We added the following phrase to the material and methods section: “The exclusion criteria were: HF de novo and age under 18 years old, inability to give informed consent.”
Comment 2. -The person who performed the echocardiography was the same for all patients?
Response 2. No, it was impossible to perform echocardiography in all patients by one doctor. Echocardiography was performed by one of few doctors working in the ward.
Comment 3. -at what point during hospitalization blood tests were taken
Response 3. Blood samples were collected during admission. We added the following phrase to the material and methods section: “The blood tests were taken on admission.“
Comment 4. -it would have been interesting to consider the background - rural or urban
Response 4. We did not gather information on this issue. We added it in limitations: “We did not gather information about the place of residence of the patients - rural or urban, or the time from HF diagnosis for all patients enrolled in this study and it was not analyzed.”
Comment 5. -it would have been interesting to consider the length of time since they have been diagnosed with heart failure
Response 5. Although it could be interesting, we did not gather data on time of HF lasting from all patients enrolled in this study and thus it was not analyzed. It could have an impact on both diuretic doses and nutritional status.
We added text to the study limitations section: “We did not gather information about the place of residence of the patients - rural or urban, or the time from HF diagnosis for all patients enrolled in this study and it was not analyzed.”
Comment 6. -how long they were treated with diuretics at the doses studied
Response 6. All patients were treated with diuretics at the studied doses for a duration of at least one month. We added the following information to the methods: “Patients received diuretics at the studied doses for at least one month.“
Discussion section
Comment 7. -comment the fact that patients with higher doses of loop diuretics were older, had comorbidities (more common AF and DM, and lower GFR) which can influent nutritional status
Response 7. We added the following text to the discussion:
“Patients with higher doses of loop diuretics were older and had more common comorbidities. It could influence the nutritional status, confounding the study results. However, propensity score matching was performed to assess whether differences in nutritional status were independent of these confounders. In matched groups of LD, MD, and HD patients (Table 3.), no differences in age and comorbidities, except for GFR were observed.”
Reviewer 2 Report
Comments and Suggestions for Authors
Dear authors, I read your article. It is well-structured, and the topic is of interest. Congratulations!
Considering that the article is well-structured and presented, a graphical abstract summarizing the study would be useful.
Regarding the materials and methods section, I would add the following notes:
The authors state that the cardiology department admitted the patients. Please mention which hospital this department belongs to.
2. The authors must clearly state the age of the patients included in the study, not just 18 years or older.
3. What is the date on which the researchers received approval from the ethics committee?
The results chapter is well-structured, the tables are easy to read, and they also contain statistically significant data. It would be useful to add some information regarding the long-term survival of the patients included in the study. What is their prognosis?
It would be intriguing to add a subsection that addresses some suggestions regarding the implementation of a diet to prevent malnutrition in these patients. How do you think these patients should be approached from a nutritional standpoint? What foods should be avoided, and what foods are recommended? You can even add a table on this topic.
Author Response
We are grateful to the Reviewers for the comments which helped us improve the manuscript. We have modified our manuscript in accordance with the recommendations of the Reviewers. We hope that now our publication will be unambiguous and more understandable to the reader.
Comments and Suggestions for Authors
Dear authors, I read your article. It is well-structured, and the topic is of interest. Congratulations!
Comment 1. Considering that the article is well-structured and presented, a graphical abstract summarizing the study would be useful.
Response 1. Graphical abstract have been already added to the submission. However now, at your request, we placed it in the manuscript file.
Regarding the materials and methods section, I would add the following notes:
Comment 2. The authors state that the cardiology department admitted the patients. Please mention which hospital this department belongs to.
Response 2. If it will be acceptable to the reviewer we would like to leave the phrase “cardiology department of university hospital”.
It was a 1st Department of Cardiology at University Hospital in Poznan, Poland
Comment 3. The authors must clearly state the age of the patients included in the study, not just 18 years or older.
Response 3. We included in the study all adult patients with HF, who were admitted to the Cardiology Department. We didn’t want to exclude patients depending on their old age. It was impossible to predict the age of the oldest patients before starting the study. Therefore, we mentioned only the lower limit of age (older than 18 years - which means adult people). We specified the age range of enrolled patients in the results: “(minimum 20, maximum 85 years)”.
Comment 4. What is the date on which the researchers received approval from the ethics committee?
Response 4. The ethics committee approval was received on April 11, 2019. It was added in the Institutional Review Board Statement: (approval code 477/19; approval date: April 11, 2019).
Comment 5. The results chapter is well-structured, the tables are easy to read, and they also contain statistically significant data. It would be useful to add some information regarding the long-term survival of the patients included in the study. What is their prognosis?
Response 5. We do not have data on patients' prognosis, because this is a cross-sectional study. We added the following phrase to limitations: “This is a cross-sectional study, consequently, we do not gather data on patients’ prognosis.”
Comment 6. It would be intriguing to add a subsection that addresses some suggestions regarding the implementation of a diet to prevent malnutrition in these patients. How do you think these patients should be approached from a nutritional standpoint? What foods should be avoided, and what foods are recommended? You can even add a table on this topic.
Response 6. The nutritional recommendations for HF patients pose a broad, important issue requiring additional publications. The European Society of Cardiology Guidelines for Diagnosis and Treatment of Heart Failure do not specify dietary intervention in this group of patients. It results probably from difficulties in leading dietary intervention studies, which require a long observational period and are prone to confounding factors. On the basis of small studies, it was assumed that HF patients should avoid salt overload (1). Well-balanced diets, including the Mediterranean diet (1,2,4) and DASH (1,3,5) may be beneficial, especially in patients with atherosclerotic aetiology of HF. Additionally some studies revealed a positive impact of supplementation of middle-chain triglyceride (MCT) oil, beta-hydroxybutyrate (BHB) salts, ketone esters, and coenzyme Q10 (CoQ10). Supplementation with these components is thought to be cardioprotective, possibly due to an increase in myocardial energy production. Additionally, supplementation with thiamine, ubiquinol, D-ribose, and L-arginine enhances left ventricular ejection fraction. Probiotic yogurt may effectively improve the inflammatory and antioxidative status of chronic heart failure. Whey protein and melatonin have a positive effect on improving endothelial function in HF patients (4). According to some studies, amino acid or protein supplementation may decrease sarcopenia and improve aerobic capacity measured as a distance in a 6-minute walk test or maximum oxygen uptake during a cardiopulmonary exercise test (6,7,8).
We added text and new citations to the introduction: “Identification of the most vulnerable patients could help to address their needs with nutritional support, including dietitian consultation. Numerous studies emphasize the importance of taking care of nutritional status in HF. Patients should avoid salt [10] and must not drink alcohol. Optimally, a balanced diet – Mediterranean [10–12] or DASH [10,13,14] should be followed. Additionally, there is data that supplementation with certain components may be beneficial: middle-chain triglyceride oil, beta-hydroxybutyrate salts, ketone esters, and coenzyme Q10 [12]. Supplementation with these components is thought to be cardioprotective, possibly due to an increase in myocardial energy production. Additionally, there are reports that supplementation with thiamine, ubiquinol, D-ribose, and L-arginine could enhance left ventricular ejection fraction [12]. Amino acid or protein supplementation may decrease sarcopenia and improve aerobic capacity [15–17].”
- Liao, L.P.; Pant, A.; Marschner, S.; Talbot, P.; Zaman, S. A Focus on Heart Failure Management through Diet and Nutrition: A Comprehensive Review. Hearts 2024, 5, 293-307. https://doi.org/10.3390/hearts5030022
- Bianchi VE. Nutrition in chronic heart failure patients: a systematic review. Heart Fail Rev. 2020;25(6):1017-1026. doi:10.1007/s10741-019-09891-1
- Yu X, Chen Q, Xu Lou I. Dietary strategies and nutritional supplements in the management of heart failure: a systematic review. Front Nutr. 2024 Oct 11;11:1428010. doi: 10.3389/fnut.2024.1428010. PMID: 39464682; PMCID: PMC11502353.
- Wickman BE, Enkhmaa B, Ridberg R, Romero E, Cadeiras M, Meyers F, Steinberg F. Dietary Management of Heart Failure: DASH Diet and Precision Nutrition Perspectives. Nutrients. 2021 Dec 10;13(12):4424. doi: 10.3390/nu13124424. PMID: 34959976; PMCID: PMC8708696.
- Levitan EB, Lewis CE, Tinker LF, Eaton CB, Ahmed A, Manson JE, Snetselaar LG, Martin LW, Trevisan M, Howard BV, Shikany JM. Mediterranean and DASH diet scores and mortality in women with heart failure: The Women's Health Initiative. Circ Heart Fail. 2013 Nov;6(6):1116-23. doi: 10.1161/CIRCHEARTFAILURE.113.000495. Epub 2013 Oct 9. PMID: 24107587; PMCID: PMC4564006.
- Aquilani R, Opasich C, Gualco A, et al. Adequate energy-protein intake is not enough to improve nutritional and metabolic status in muscle-depleted patients with chronic heart failure. Eur J Heart Fail. 2008;10(11):1127-1135. doi:10.1016/j.ejheart.2008.09.002
- Deutz NE, Matheson EM, Matarese LE, Luo M, Baggs GE, Nelson JL, Hegazi RA, Tappenden KA, Ziegler TR; NOURISH Study Group. Readmission and mortality in malnourished, older, hospitalized adults treated with a specialized oral nutritional supplement: A randomized clinical trial. Clin Nutr. 2016 Feb;35(1):18-26. doi: 10.1016/j.clnu.2015.12.010. Epub 2016 Jan 18. PMID: 26797412.
- Lombardi C, Carubelli V, Lazzarini V, Vizzardi E, Quinzani F, Guidetti F, Rovetta R, Nodari S, Gheorghiade M, Metra M. Effects of oral amino Acid supplements on functional capacity in patients with chronic heart failure. Clin Med Insights Cardiol. 2014 May 21;8:39-44. doi: 10.4137/CMC.S14016. PMID: 24899826; PMCID: PMC4039212.
Reviewer 3 Report
Comments and Suggestions for Authors
The authors describe the work on exploring the relationship between the use of high-dose loop diuretics and nutritional status in patients with HFrEF. In these 353 hospitalized patients with HFrEF study, it was found the mean MNA score was 23.31 ± 2.93 points, and 49.8% were at risk of malnutrition or were malnourished. According to the MNA and the GNRI, patients in high dose (HD) and medium dose (MD) groups had worse nutritional status than the low dose (LD) group. For CONUT, the differences were significant between all groups: nutritional status was the worst in the HD group, intermediate in the MD group, and the best in the LD group. The authors concluded that the intake of loop diuretics, especially in high doses, correlates with elevated risk of malnutrition in patients with HFrEF independently of sex, age, NYHA class and left ventricular ejection fraction. This is an interesting study. Appropriate methodology has been employed and the conclusions appear to be justified based on the data. The authors describe some of the limitations of their work. However, I have some recommendations for consideration.
- It would be helpful if the authors can provide a stronger rationale for why the study was undertaken, as well as a provide a clear hypothesis to be tested in the study?
- Results/Discussion. Are the authors able to indicate which specific nutrients may be deficient in patients with HFrEF and on HD loop diuretics?
- Is there a possibility that switching patients from HD to MD/LD dose improves nutritional status?
- With respect to effects of HD diuretics in this patient population, is this a specific effect in patients with HFrEF or would they also be expected to influence nutritional status in patients with other pathophysiological conditions such as hypertension, and renal disease?
- Can the authors elaborate and emphasize the clinical/nutritional application of their findings?
- Please review manuscript for English language grammatical errors.
Comments on the Quality of English Language
Please review manuscript for English language grammatical errors.
Author Response
We are grateful to the Reviewers for the comments which helped us improve the manuscript. We have modified our manuscript in accordance with the recommendations of the Reviewers. We hope that now our publication will be unambiguous and more understandable to the reader.
The authors describe the work on exploring the relationship between the use of high-dose loop diuretics and nutritional status in patients with HFrEF. In these 353 hospitalized patients with HFrEF study, it was found the mean MNA score was 23.31 ± 2.93 points, and 49.8% were at risk of malnutrition or were malnourished. According to the MNA and the GNRI, patients in high dose (HD) and medium dose (MD) groups had worse nutritional status than the low dose (LD) group. For CONUT, the differences were significant between all groups: nutritional status was the worst in the HD group, intermediate in the MD group, and the best in the LD group. The authors concluded that the intake of loop diuretics, especially in high doses, correlates with elevated risk of malnutrition in patients with HFrEF independently of sex, age, NYHA class and left ventricular ejection fraction. This is an interesting study. Appropriate methodology has been employed and the conclusions appear to be justified based on the data. The authors describe some of the limitations of their work. However, I have some recommendations for consideration.
Comment 1. It would be helpful if the authors can provide a stronger rationale for why the study was undertaken, as well as a provide a clear hypothesis to be tested in the study?
Response 1. In this study the following hypothesis was formulated: In patients with heart failure with reduced ejection fraction, is there a correlation between the dosage of loop diuretics and nutritional status? First of all, the study indicates that patients with HFrEF with high doses of diuretics should undergo regular monitoring of their nutritional status and possible prevention of malnutrition. It is not possible to conclude on the basis of this study about the causal-effect relationship between the dose of diuretics and malnutrition. However, the higher incidence of malnutrition in patients taking higher doses of loop diuretics may suggest a special need for dietary intervention in this group of patients (possible recommendations were added in the introduction on another Reviewer's request). On the other hand, some patients receive but do not require high doses of diuretics, and their doses can be carefully reduced with regular monitoring of euvolemia (22).
- Rohde LE, Rover MM, Figueiredo Neto JA, et al. Short-term diuretic withdrawal in stable outpatients with mild heart failure and no fluid retention receiving optimal therapy: a double-blind, multicentre, randomized trial. Eur Heart J. 2019;40(44):3605-3612. doi:10.1093/eurheartj/ehz554
We added text to the introduction to provide stronger rationale:
“The study results could help select especially vulnerable patients who should undergo more frequent monitoring of nutritional status and nutritional interventions.”
“It was done to assess if there is a correlation between the dosage of loop diuretics and nutritional status.”
Comment 2. Results/Discussion. Are the authors able to indicate which specific nutrients may be deficient in patients with HFrEF and on HD loop diuretics?
Response 2. At your request, the following text and citations were added to the discussion:
“Loop diuretic use can cause a deficit of nutrients in two ways: causing a loss of electrolytes with urine (33-35) and additionally may affect appetite and therefore increase the risk of malnutrition. Loop diuretics cause a substantial loss of both magnesium and potassium in the plasma and intracellular space (33,34). Increasing doses of diuretics can compromise renal perfusion, stimulating the renin-angiotensin-aldosterone system. Simultaneous activation of vasopressin can cause retention of free water, leading to hyponatremia (35). On the other hand, diuretics can lead to decreased appetite by changing the composition of saliva. Impaired appetite can lead to reduced food intake and, consequently, malnutrition, causing a deficiency of various nutrients (36).”
- Dunn SP, Bleske B, Dorsch M, Macaulay T, Van Tassell B, Vardeny O. Nutrition and heart failure: impact of drug therapies and management strategies. Nutr Clin Pract. 2009 Feb-Mar;24(1):60-75. doi: 10.1177/0884533608329299. PMID: 19244150.
- Schwinger RH, Erdmann E. Heart failure and electrolyte disturbances. Methods Find Exp Clin Pharmacol. 1992 May;14(4):315-25. PMID: 1507935.
- Nicholls MG. Interaction of diuretics and electrolytes in congestive heart failure. Am J Cardiol. 1990 Mar 6;65(10):17E-21E; discussion 22E-23E. doi: 10.1016/0002-9149(90)90246-w. PMID: 2309624.
- Rothberg MB, Sivalingam SK. The new heart failure diet: less salt restriction, more micronutrients. Journal of General Internal Medicine. 2010 Oct;25(10):1136-1137. DOI: 10.1007/s11606-010-1254-8. PMID: 20151221; PMCID: PMC2955483.
Comment 3. Is there a possibility that switching patients from HD to MD/LD dose improves nutritional status?
Response 3. Some patients with HFrEF do not require high doses of diuretics, and their doses can be carefully reduced under regular monitoring of euvolemia (22). Usually, higher doses are given after discharge from the hospital (de novo diagnosis, exacerbation of HF), and then they are reduced. Considering the pathophysiological basis: the effect of diuretics on the sense of taste - metallic taste as one of their side effects, reducing the dose of diuretics may be helpful in improving appetite and consequently the nutritional status.
Following text was added to introduction
“Some patients with HFrEF do not require high doses of diuretics, and their doses can be carefully reduced under regular monitoring of euvolemia (22). Usually, higher doses are given during the hospitalization or close after the discharge from the hospital (de novo diagnosis, exacerbation of HF) and then reduced. As loop diuretics could cause a metallic taste as one of their side effects, reducing the dose of diuretics may theoretically improve appetite and consequently the nutritional status.”
- Rohde LE, Rover MM, Figueiredo Neto JA, et al. Short-term diuretic withdrawal in stable outpatients with mild heart failure and no fluid retention receiving optimal therapy: a double-blind, multicentre, randomized trial. Eur Heart J. 2019;40(44):3605-3612. doi:10.1093/eurheartj/ehz554
Comment 4. With respect to the effects of HD diuretics in this patient population, is this a specific effect in patients with HFrEF, or would they also be expected to influence nutritional status in patients with other pathophysiological conditions such as hypertension, and renal disease?
Response 4. Patients with hypertension are rarely prescribed high doses of loop diuretics as loop diuretics are not the first line of antihypertensive drugs as well and are usually not escalated to high doses until hypertension does not coexist with heart failure. This study was based on HFrEF patients, and we think it should not be generalized to other diseases like hypertension or renal disease. Maybe similar results could be gathered in patients with HF with LVEF>40% (HFmrEF and HFpEF).
Comment 5. Can the authors elaborate and emphasize the clinical/nutritional application of their findings?
Response 5. Firstly, it suggests that patients with HFrEF who are taking high doses of diuretics should undergo regular monitoring of their nutritional status and possible prevention of malnutrition. It is not possible to conclude on the basis of this study about the causal-effect relationship between the dose of diuretics and malnutrition. Nevertheless, conducting an experimental study seems ethically questionable. On the other hand, some patients receive but do not require high doses of diuretics, and their doses can be carefully reduced (22) with regular monitoring of euvolemia.
We added text to the discussion:
“Previous research indicates that loop diuretics doses can be reduced in stable HFrEF patients [22]. Our results may indicate that such a reduction in loop diuretic use could have a beneficial effect on nutritional status, however it requires further experimental studies.”
- Rohde LE, Rover MM, Figueiredo Neto JA, et al. Short-term diuretic withdrawal in stable outpatients with mild heart failure and no fluid retention receiving optimal therapy: a double-blind, multicentre, randomized trial. Eur Heart J. 2019;40(44):3605-3612. doi:10.1093/eurheartj/ehz554
Comment 6. Please review manuscript for English language grammatical errors.
Response 6. We carefully checked the manuscript and corrected grammatical errors through manuscript.
Round 2
Reviewer 1 Report
Comments and Suggestions for Authors
The authors have responded to all suggestions made. The manuscript is improved and deserves to be published.
Reviewer 2 Report
Comments and Suggestions for Authors
The authors fulfilled all the requests.
The paper can be published in the presented form.